# UNSUPERVISED PRIOR LEARNING: DISCOVERING CATEGORICAL POSE PRIORS FROM VIDEOS

## ABSTRACT

A prior represents a set of beliefs or assumptions about a system, aiding inference and decision-making. In this work, we introduce the challenge of unsupervised prior learning in pose estimation, where AI models learn pose priors of animate objects from videos in a self-supervised manner. These videos present objects performing various actions, providing crucial information about their keypoints and connectivity. While priors are effective in pose estimation, acquiring them can be difficult. We propose a novel method, named Pose Prior Learner (PPL), to learn general pose priors applicable to any object category. PPL uses a hierarchical memory to store compositional parts of prototypical poses, from which we distill a general pose prior. This prior enhances pose estimation accuracy through template transformation and image reconstruction. PPL learns meaningful pose priors without any additional human annotations or interventions, outperforming competitive baselines on both human and animal pose estimation datasets. Notably, our experimental results reveal the effectiveness of PPL using learnt priors for pose estimation on occluded images. Through iterative inference, PPL leverages priors to refine estimated poses, regressing them to any prototypical poses stored in memory. Our code, model, and data will be publicly available.

## 1 INTRODUCTION

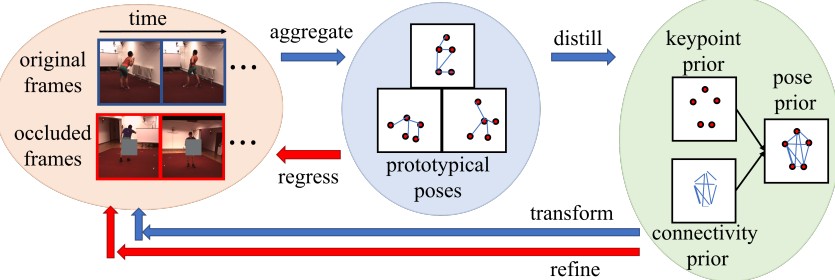

Figure 1: **Schematic illustration of the unsupervised prior learning challenge on videos in the task of pose estimation.** Given a series of original video frames (framed in blue), the challenge is to learn the pose prior (green circle) in a self-supervised manner. The pose prior comprises keypoint and connectivity priors. To address this challenge, we propose a Pose Prior Learner (PPL). During PPL training, prototypical poses in the memory (blue circle) are aggregated from the individual poses estimated from all video frames (orange circle) and distilled into a general pose prior. The pose prior can then guide pose estimation from video frames through transformations. This cyclic process strengthens the learning of robust pose prior representations, resulting in more accurate pose estimations, which, in turn, helps capture more representative prototypical poses. During inference, the learned pose prior refines pose estimation on occluded images (bounded in red), regressing them to the prototypical poses stored in memory. Blue arrows illustrate signal flows during PPL training, while red arrows indicate signal flows during inference on occluded images.

Priors represent beliefs or assumptions about a system or the characteristics of a concept. They are widely used in statistical inference (Lindley, 1961), cognitive science (Schad et al., 2021), and

machine learning (Diligenti et al., 2017; Gülçehre & Bengio, 2016). This pre-existing knowledge is essential for guiding the inference process, enabling AI models to make robust predictions in uncertain or ambiguous situations (Thiruvenkadam et al., 2008; Sung et al., 2015; Liang et al., 2024). The objective of our work is to enhance our understanding of priors in AI models and offer preliminary answers to the three key intelligence questions: (1) How do we acquire priors in the first place? (2) Can we learn them from input data in a self-supervised manner? (3) Can we evaluate and enhance the quality of the priors? To tackle these questions, we first introduce the challenge of unsupervised prior learning in the context of pose estimation from videos. See **Figure 1** for the schematic illustration of the challenge. Pose estimation is a classical computer vision task that identifies the structure of objects in images and videos by detecting keypoints. A pose prior summarizes the common characteristics shared by a variety of poses. It encapsulates the expectation of the keypoint configurations and the connectivity between keypoints.

In parallel to our challenge of unsupervised prior learning from videos for pose estimation, unsupervised pose estimation leverages the abundant, unannotated visual information available in large image and video datasets to extract pose information (Hu & Ahuja, 2021; Sommer et al., 2024; Chen et al., 2019; He et al., 2022a; Schmidtke et al., 2021). The use of pose priors can provide valuable guidance in this process. We categorize the existing works into two groups: those that incorporate priors and those that operate without them.

Recent approaches (He et al., 2022a; Sun et al., 2022; 2023) attempt to predict keypoints from images, construct object structure representations using these keypoints, and learn effective structural information through image reconstruction. However, without pose priors, these methods can be easily disrupted by background information, causing the model to occasionally learn the background structure instead of the foreground object. While using video streams with static backgrounds helps alleviate this issue (Schmidtke et al., 2021; Yoo & Russakovsky, 2023), these methods remain vulnerable to producing inaccurate keypoints. This vulnerability arises from the lack of supplementary knowledge that could constrain both keypoint localization and the connectivity among keypoints.

In contrast, some methods (Schmidtke et al., 2021; Yoo & Russakovsky, 2023) utilize prior knowledge of a category's pose to guide the pose estimation of individuals within that category. Conceptually, each category is expected to exhibit a generalized and distinctive pose prior that reflects characteristics such as shape, size, and structure. Individual poses should be seen as geometric transformations of this category-specific pose prior. As a result, employing a category-specific pose prior aids in guiding and regularizing the learning of poses. However, acquiring comprehensive general pose priors is highly challenging, as it necessitates extensive human annotations and domain knowledge.

Loosely inspired by how humans develop a universal prior representation of an object category by observing the movements of individual object instances in videos and subsequently using these priors to estimate individual poses, we propose a new method called the Pose Prior Learner (PPL). PPL is designed to effectively learn a meaningful pose prior for any object category. It utilizes a hierarchical memory to store a finite set of prototypical poses and extract a general pose prior from them. Initially, both the hierarchical memory and the universal prior representation are randomly initialized but learnable parameters. During training, effective pose learning is ensured through image reconstruction. As training progresses, the hierarchical memory retains and aggregates multiple accurate prototypical poses, thereby contributing to a more precise pose prior and enhancing the model's ability to estimate poses.

Upon completing the training, we obtain a model that enables accurate pose estimation, a pose prior that encapsulates the general features of a category, and a hierarchical memory that stores diverse prototypical poses for that category. We evaluate the effectiveness of our PPL across several human and dog pose estimation benchmarks. We visualize their pose priors to further interpret what our approach has learned. Additionally, we introduce an iterative inference strategy to estimate the poses of objects in occluded scenes using the trained hierarchical memory and the pose prior. Our contributions are highlighted below:

**1.** We introduce the challenge of unsupervised prior learning in the context of pose estimation. We also establish evaluation metrics and benchmarks to assess the quality of the learned priors.

**2.** We propose a new method called Pose Prior Learner (PPL) for unsupervised pose estimation. PPL effectively mitigates background noise and does not require any domain knowledge from humans. The compositional hierarchical memory in PPL aggregates prototypical poses from training videos and subsequently utilizes these poses to distill pose priors. Three training techniques are introduced to ensure training convergence and stability of PPL.

**3.** PPL outperforms existing methods across several pose estimation benchmarks and offers explainable visualizations of pose priors. Notably, We found that predefined human priors are not always optimal. Our PPL even outperforms models using human-defined priors.

**4.** During inference, we utilize an iterative strategy in which PPL progressively leverages priors to refine estimated poses by regressing them to the nearest prototypical poses stored in memory. Experimental results demonstrate that our PPL accurately estimates poses, even in occluded scenes.

## 2 RELATED WORKS

**Unsupervised Pose Estimation without Priors.** Numerous methods without priors have been proposed to detect keypoints from images or videos, which are then used to reconstruct images for supervision (Li et al., 2021; Geng et al., 2021; Zhang et al., 2018; Sun et al., 2022; Thewlis et al., 2017; Jakab et al., 2020). For example, AutoLink (He et al., 2022a) extracts keypoints from the image and estimates the strength of the links between pairs of keypoints. It then combines these keypoints with the link heatmap to reconstruct the randomly masked image. Similarly, BKind (Sun et al., 2022; 2023) uses keypoints extracted from two video frames to reconstruct the pixel-level differences between these two frames. In these methods, keypoints are directly predicted from the image and supervised solely by image reconstruction, leading to potential detection of keypoints in background regions with complex textures. Moreover, the absence of constraints on keypoint configuration and connectivity also contributes to the unreliability of their methods. In contrast, our PPL utilizes the learned pose prior as a constraint to mitigate these issues.

**Unsupervised Pose Estimation Incorporating Priors.** Several methods utilize prior knowledge to guide the pose estimation (Chen & Dou, 2021; Shi et al., 2023; Zhang et al., 2022; Schmidtke et al., 2021; Yoo & Russakovsky, 2023). Among these methods, Shape Template Transforming (STT) (Schmidtke et al., 2021) applies affine transformations to a predefined pose prior, aligning it with the estimated pose from a video frame. By incorporating an additional frame from the same video to provide background information, an image reconstruction loss supervises the pose estimation process. The pose prior effectively guides pose estimation by constraining the shape of the human pose and the connectivity between body parts. However, pose priors are often difficult to obtain, requiring costly human annotations. Moreover, HPE (Yoo & Russakovsky, 2023) has shown that predefined pose priors are not always optimal, and tuning the shape of the prior can sometimes improve performance. Unlike these methods, our approach learns the prior directly from input videos, and models with our learned priors even outperform those using human-defined priors.

**Compositional Memory Architectures.** Compositional memory has been widely used in many computer vision tasks, such as question answering (Seong et al., 2021), object segmentation (Seong et al., 2021), and sence graph generation (Deng et al., 2022). In pose estimation, PCT (Geng et al., 2023) decomposes a human pose into discrete tokens, where each token connects several interdependent joints and characterizes a sub-structure of the entire human pose. This approach is highly effective for decomposing and reconstructing poses, providing robust pose representations while significantly reducing computation and storage costs. However, PCT encodes all tokens into the same embedding space, making it difficult to aggregate semantic tokens that represent different sub-structures. In contrast, our PPL employs a compositional hierarchical memory, which parses poses into memory banks. Each memory bank explicitly contains multiple tokens encoding the variations of each sub-structure of a pose. This facilitates the aggregation of poses into a universal pose prior.

## 3 OUR PROPOSED METHOD – POSE PRIOR LEARNER (PPL)

We introduce our proposed method, Pose Prior Learner (PPL). Given videos with static backgrounds featuring object instances from a specific category, such as dogs or humans, PPL can accurately

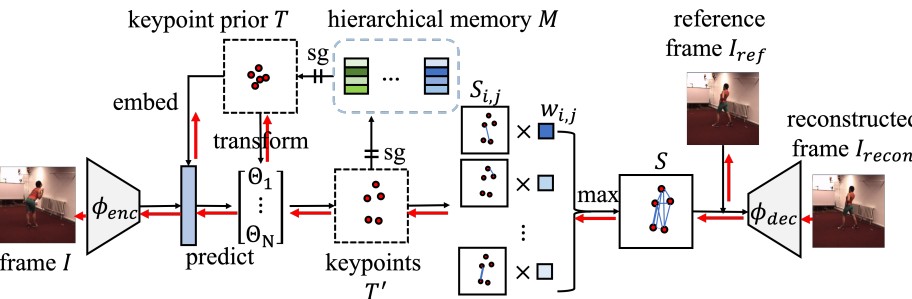

Figure 2: **Overview of our proposed Pose Prior Learner (PPL).** We first distill the keypoint prior from the hierarchical memory $M$. Features of the frame $I$ and the embedding of the keypoint prior are concatenated to predict the affine transformation parameters. The keypoint prior is transformed and their pair-wise links are modulated with the connectivity prior $W$ to obtain the combined link heatmap $S$. The concatenation of the link heatmap $S$ and the reference frame $I_{ref}$ is decoded to produce the reconstructed frame $I_{recon}$. The $sg$ symbol represents the stopping gradient operation. The red arrows indicate the gradient flows during backpropagation based on image reconstruction. See **Section 3.4** for training details.

estimate the poses of the objects in that category while gradually learning a general pose prior through unsupervised learning. Note that our PPL requires no extra knowledge from human annotators. The architecture of PPL is presented in **Figure 2**.

Mathematically, We represent the topology for an object as a graph connecting keypoints with shared link weights, also known as connectivity. For any category, its general pose prior $V$ is defined as $V = (T, W)$, where $T$ represents its keypoint prior and $W$ denotes the connectivity prior. Specifically, $T$ consists of $N$ keypoints: $T = [P_1, P_2, ..., P_N]$, where $P_i \in [-1, 1] \times [-1, 1]$ is the normalized pixel coordinates. $W$ is a $2D$ matrix of size $N \times N$, where each entry $w_{ij}$ in the matrix represents the connectivity probability between two keypoints $P_i$ and $P_j$. For instance, in the case of humans, the hand is connected to the torso via an arm; thus, the connectivity probability between these two parts should be higher than the connectivity probability between a hand and a foot. We randomly initialize the connectivity prior $W$ with positive values. The keypoint prior $T$ is decoded from a hierarchical memory $M$ storing compositional parts of prototypical poses.

During training, PPL inputs a pair of frames $I$ and $I_{ref}$ with the same sizes of H × W × 3 where H and W are the height and the width of each video frame. The reference frame $I_{ref}$ is a randomly selected frame from the same video, containing the same person as $I$ but in a different pose. The aim of PPL is to learn to correctly predict the keypoints $T'$ and their connectivity on $I$. Ideally, if PPL makes perfect predictions of the pose on $I$, by combining it with the background information from $I_{ref}$, the reconstructed image $I_{recon}$ should match $I$ exactly. Next, we introduce how we estimate $T'$ on $I$ using the keypoint prior $T$.

## 3.1 TRANSFORMATION OF THE KEYPOINT PRIOR

Given the frame $I$, we use a feature extractor $\phi_{enc}$ to extract its embedding $h_I$: $h_I = \phi_{enc}(I)$. $\phi_{enc}$ is a 2D-Convolution Neural Network (2D-CNN) trained from scratch. The keypoint prior $T$ is converted into an embedding $h_T$ via a series of fully connected layers. Together with $h_T$ as inputs, PPL learns to predict the affine transformation parameters $\Theta_i \in [\Theta_1, \Theta_2, ..., \Theta_N]$ for each keypoint $P_i$ in $T$ from $h_I$ via a two-layer fully connected network denoted as $FC(\cdot)$:

$$[\Theta_1, \Theta_2, ..., \Theta_i, ..., \Theta_N] = FC(h_I, h_T), \text{ where } \Theta_i = \begin{bmatrix} a^{(i)} & b^{(i)} & t_x^{(i)} \\ c^{(i)} & d^{(i)} & t_y^{(i)} \\ 0 & 0 & 1 \end{bmatrix}, \quad (1)$$

where $t_x^{(i)}$ and $t_y^{(i)}$ are the translations and $a^{(i)}, b^{(i)}, c^{(i)}, d^{(i)}$ are the coefficients that define rotation, scaling, and shear. Each point $P_i$ in $T$ can then be transformed by $\Theta_i$, resulting in the keypoints $T'$ for the frame $I$:

$$T' = [P_1', P_2', ..., P_N'], \text{ where } [P_i', 1]^\top = \Theta_i [P_i, 1]^\top. \quad (2)$$

## 3.2 Connecting keypoints based on the connectivity prior

The connectivity of keypoints in objects is often fixed and rigid; for example, human arms maintain a relatively constant length, with a hand always connected to the torso via an arm and never connected to a foot. This rigidity in connectivity serves as a constraint, aiding in the regularization of pose estimation. In this section, we introduce the connectivity prior and explain how it can be used to regularize the connectivity strength between any pair of estimated keypoints in $T'$ on $I$.

Similar to AutoLink (He et al., 2022a), PPL connects any two keypoints $P'_i$ and $P'_j$ in $T'$ to obtain differentiable link heatmap $S_{i,j} \in \mathbb{R}^{H \times W}$. Intuitively, each 2D link heatmap represents a probability density map, where the pixel values along the link between two points are high, while other areas are assigned values close to zero. For any point $P'_i$, its strongest connectivity to any of the other points in $T'$ is activated on the combined link heatmap $S \in \mathbb{R}^{H \times W}$ via a max pooling operation over all the $N \times N$ link heatmaps:

$$S = \max_{i,j}^{N \times N}(w_{i,j}S_{i,j}), \tag{3}$$

where $w_{i,j}$ in the connectivity prior $W$ modulates the link heatmap $S_{i,j}$ based on whether the two keypoints $P'_i$ and $P'_j$ are physically connected. Ideally, if PPL correctly estimates the probability of physical links for an object category, $S_{i,j}$ will receive higher connectivity values, thereby activating the locations linking these two keypoints on the combined link map $S$.

Given the combined link map $S$ and the reference frame $I_{ref}$, PPL can reconstruct the image $I$. $I_{ref}$ shares the same background as $I$ but features a different foreground pose, thereby offering background information for reconstruction. Meanwhile, $S$ provides the connectivity details among all the estimated keypoints. This encourages PPL to learn accurate pose estimations while minimizing the influence of background noise. Therefore, we concatenate $S$ and $I_{ref}$ and feed them into a 2D-CNN to perform the image reconstruction $I_{recon}$, where $I_{recon} = \phi_{dec}(I_{ref}, S)$.

## 3.3 Reconstruction of Keypoint Configuration with Memory

$M$ is a hierarchical memory storing compositional representations of prototypical poses. It contains $m$ memory banks $\{b_1, b_2, ..., b_m\}$ of $k$ learnable tokens with $d$ randomly initialized parameters per token. $M$ is important to learn robust prior representations $V$ due to the following reasons: (1) By storing multiple prototypical poses, memory helps in aggregating these poses, which aids in creating a more robust and comprehensive prior that captures variations within an object category. (2) $M$ can assist in organizing information hierarchically, making it easier for PPL to retrieve relevant prototypical poses when making predictions in uncertain or ambiguous scenarios, such as occlusion. and (3) By leveraging $M$, PPL can refine its predictions iteratively, using stored poses to adjust its outputs based on previously learned pose representations. Next, we introduce how $M$ is structured.

Given the estimated $N$ keypoints in $T'$, we first encode it into $m$ tokens of dimension $d$ with several MLP-Mixer blocks (Tolstikhin et al., 2021), denoted as $MIX_{enc}$. We define $G$ as the collection of the $m$ tokens, where each token $g_i$ represents the embedding of a subset of the keypoints. Each token $g_i$ shares the same embedding space with the memory bank $b_i$ in memory $M$:

$$G = [g_1, g_2, ..., g_m] = MIX_{enc}(T'), \tag{4}$$

If all the tokens in all the memory banks of $M$ learn to capture unique parts of prototypical poses, each $g_i$ should be able to retrieve the most similar token in the memory bank $b_i$ of $M$ and reconstructs $G$ itself. Here, we use $L2$ distance to measure the similarity between $g_i$ and $k$ token in $b_i$ of $M$. The token in $b_i$ that is most similar to $g_i$ is denoted as $g'_i$. Thus, for each $g_i$, we can always find the corresponding $g'_i$ from $b_i$, resulting in a collection of $G' = [g'_1, g'_2, ..., g'_m]$. Another series of MLP-Mixer blocks (denoted as $MIX_{dec}$) is then used to decode $G'$ back to $N$ keypoints $T'_{recon}$, where $T'_{recon} = MIX_{dec}(G')$. See **Figure 3(a)** for the schematic.

Unlike PCT (Geng et al., 2023) where all tokens of a memory are within the same embedding space, We organize $M$ into $m$ memory banks, each containing $k$ tokens that represent an independent embedding space. This structure enables us to efficiently distill compositional parts from each memory bank, allowing us to form a general representation of a pose, which is distilled into the keypoint prior $T$. Specifically, we define $MP(\cdot)$ as the mean pooling operation. PPL pools one

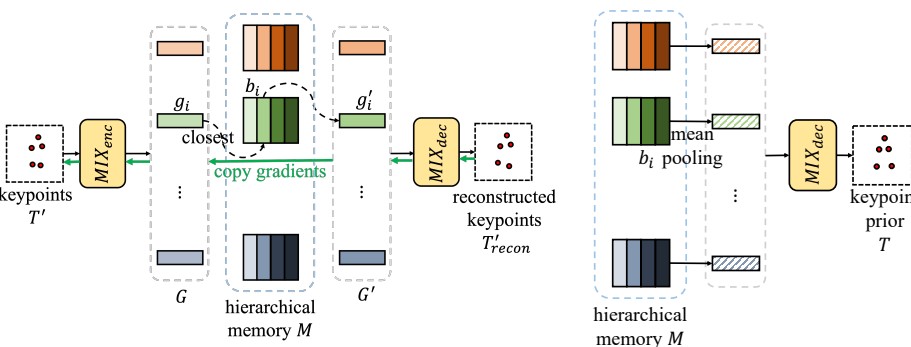

(a) Keypoint configuration reconstruction.  (b) Memory distillation.

Figure 3: **Retrieval and Distillation of the proposed Hierarchical Memory in our PPL.** (a) The hierarchical memory $M$ is trained to reconstruct the keypoints $T'_{recon}$. $T'$ is encoded into $m$ tokens by the MLP-Mixer blocks $MIX_{enc}$. Each token $g_i$ retrieves its closest token $g'_i$ in memory bank $b_i$. The resulting $m$ tokens are decoded by the MLP-Mixer $MIX_{dec}$ into the reconstructed keypoints $T'_{recon}$. The green arrows indicate the gradient flows during backpropagation based on the reconstruction of keypoint configurations. See **Section 3.4** for training details. (b) The hierarchical memory $M$ is distilled into the keypoint prior $T$. Vectors in every memory bank $b_i$ are mean-pooled into one vector, and the resulting $m$ vectors are decoded by $MIX_{dec}$ into the keypoint prior $T$. See **Section 3.3** for details.

token vector among the $k$ vectors of each memory bank. These pooled $m$ vectors are further decoded by $MIX_{dec}$ into $N$ points of our distilled keypoint prior $T$. See **Figure 3(b)** for the schematic of distilling $T$: $T = MIX_{dec}([MP(b_1), MP(b_2), ..., MP(b_m)])$.

### 3.4 TRAINING AND INFERENCE

Our PPL is trained to jointly minimize all the four losses: the image reconstruction loss $L_{ir}$, the boundary loss $L_b$, the link regularization loss $L_l$, and the keypoint configuration reconstruction loss $L_{kr}$. We elaborate on these four losses below.

**Image Reconstruction Loss.** If PPL correctly estimates the pose on the original image $I$, the reconstructed image $I_{recon}$, based on the estimated pose, should be identical to $I$. Therefore, ensuring the quality of $I_{recon}$ encourages PPL to improve its pose estimation accuracy. To achieve this, we apply a perceptual loss on the embeddings of $I$ and $I_{recon}$, extracted using a frozen feature extractor $\psi(\cdot)$ from the VGG19 network pre-trained on ImageNet (Russakovsky et al., 2015). The perceptual loss is defined as: $L_{ir} = \|\psi(I_{recon}) - \psi(I)\|_1$.

**Boundary Loss.** To ensure that the network does not transform the points in the keypoint prior outside the boundaries of the image, we limit the x and y coordinates of the transformed keypoints to be within the image:

$$L_b = \sum_{* \in x,y} \begin{cases} |P'_{i,*}| & \text{if } |P'_{i,*}| > 1, \\ 0 & \text{otherwise.} \end{cases} \text{ and } L_l = \sum_{i,j} w_{i,j} \|l(P_i, P_j) - l(P'_i, P'_j)\|_1, \qquad (5)$$

where $P'_{i,x}$ and $P'_{i,y}$ are the normalized $x$ and $y$ coordinate of the keypoint $P'_i$ respectively.

**Link Regularization Loss.** A person's arm always maintains a fixed length regardless of the poses. Thus, we propose the constraint that links should be assigned a high weight if they do not vary significantly in length before and after the affine transformation. The loss $L_l$ encourages the preservation of link lengths during pose estimation. It is defined as in the equation above, where $l(\cdot)$ is the L2 distance between two keypoints before and after the affine transformation.

**Reconstruction Loss on Keypoint Configurations.** In **Section 3.3**, given a collection of token representations in $G$, PPL retrieves the most similar tokens from each memory bank of the hierarchical memory $M$ and generates $G'$ in a non-differentiable manner. To ensure that $M$ learns to store meaningful token embeddings that represent compositional parts of poses, the retrieved

Figure 4: **Overview of the iterative inference strategy in our PPL.** During inference, we iteratively use the reconstructed frame $I_{recon}$ as input to estimate the pose $T'$. The hierarchical memory $M$ refines the estimated pose $T'$ and outputs $T'_{recon}$. With the original test image $I$ as the reference frame $I_{ref}$, PPL reconstructs the image $I_{recon}$. It is then used as the input frame in the next iteration. See **Section 3.4** for details.

token representations $G'$ from $M$ should closely match $G$. Moreover, if these compositional parts are structured correctly, the tokens should be able to decode into meaningful keypoint configurations $T'_{recon}$ that are close to the original keypoint configurations $T'$. Therefore, we introduce the keypoint configuration reconstruction loss $Lkr$, defined as: $L_{kr} = \|T'_{recon} - T'\|_2 + \|G - G'\|_2$.

**Training Techniques.** To ensure convergence and stability during the training of PPL, we introduce three gradient dettachment techniques: (1) To address the broken gradient issue during the quantization step from $G$ to $G'$, we adopt the approach from VQ-VAE (Van Den Oord et al., 2017). Specifically, our PPL copies the gradients of $G'$ to $G$ for backward propagation, allowing the gradients to flow through the quantization step. (2) The hierarchical memory $M$ is updated using an exponential moving average to smooth the gradient updates, particularly during the early stages of training when $G$ can be quite noisy. This approach helps stabilize the learning process and ensures that $M$ retains more reliable information over time. (3) For $M$ to learn effective representations of $G$, it requires an accurate estimation of $T'$, which depends on a good prior $V$ distilled from $M$. This creates a chicken-and-egg problem that complicates training. To address this, we introduce two gradient detachments to separate the training processes. First, we detach the gradients from $T$ and train the keypoint transformation and image reconstruction pathway, as shown by the red arrows in **Figure 2**. Second, we detach the gradients from $T'$ to train the memory encoder and decoder, $MIX_{enc}$ and $MIX_{dec}$, as indicated by the green arrows in **Figure 3(a)**.

**Iterative Inference.** We propose an iterative inference strategy (**Figure 4**). 4 iterations are used for every experiment. In every iteration, we take the reconstructed frame $I_{recon}$ from the last iteration (the original frame $I$ for iteration 0) as the input. We infer its keypoints $T'$ as the output keypoints of the current iteration. The hierarchical memory $M$ is used to reconstruct $T'$ and the reconstructed keypoints $T'_{recon}$ are used to obtain the reconstructed frame $I_{recon}$. $I_{recon}$ is then used as the input for the next iteration. We keep the original occluded frame $I$ as the reference frame for all the iterations. See **Appendix A1** for more implementation details.

## 4 EXPERIMENTS

We use three video datasets: Human3.6m (Ionescu et al., 2013), Taichi (Siarohin et al., 2019), and YouTube dog videos with green backgrounds for our experiments. See **Appendix A2** for details. On Human3.6m, we report the results in the mean $L2$ error between the predicted keypoints and the ground truth, normalized by the image size. For Taichi, as in (He et al., 2022a; Siarohin et al., 2021; He et al., 2021; 2022b; Zhang et al., 2018) we use the summed $L2$ error computed at a resolution of $256 \times 256$. To ensure a more accurate evaluation, unlike previous methods (He et al., 2022a; Schmidtke et al., 2021), we employ a minimum flow algorithm (Waissi, 1994) to align the predicted keypoints with the ground truth.

### 4.1 UNSUPERVISED HUMAN POSE ESTIMATION

We compare the keypoint detection results of PPL with other unsupervised pose estimation methods and present the results in **Table 1** (Human3.6m) and **Table 2** (Taichi). On both datasets, PPL significantly outperforms all baselines across all image resolutions. Among the baselines, AutoLink (He et al., 2022a) also incorporates learnable connectivity priors, similar to our PPL. However, its performance is inferior due to the absence of hierarchical memory. Additionally, we note that STT

Table 1: **Keypoint detection on Human3.6m.** We report the mean $L2$ error normalized by the image resolutions (Res.) of both $256 \times 256$ and $128 \times 128$. $*$ is the $L2$ error calculated by our minimum flow algorithm. The best is in bold.

| Method | Res. | Norm. $L2$ Error |
|---|---|---|
| (Jakab et al., 2020) | 256 | 2.73 |
| (Thewlis et al., 2017) | 256 | 7.51 |
| (Lorenz et al., 2019) | 256 | 2.79 |
| (Zhang et al., 2018) | 256 | 4.91 |
| (Schmidtke et al., 2021) | 256 | 3.31 |
| AutoLink (He et al., 2022a) | 128 | 2.76 |
| AutoLink* (He et al., 2022a) | 128 | 2.61 |
| PPL (ours) | 128 | **1.92** |
| PPL (ours) | 256 | **2.56** |

Table 2: **Keypoint detection on Taichi.** For consistency with the baseline methods, we report the summed $L2$ error at a resolution of $256 \times 256$. $*$ is the $L2$ error calculated by our minimum flow algorithm. The best is in bold.

| Method | Summed $L2$ Error |
|---|---|
| (Siarohin et al., 2021) | 389.78 |
| (He et al., 2021) | 437.69 |
| (Zhang et al., 2018) | 343.67 |
| (He et al., 2022b) | 417.17 |
| AutoLink (He et al., 2022a) | 316.10 |
| AutoLink* (He et al., 2022a) | 302.41 |
| PPL (ours) | **293.35** |

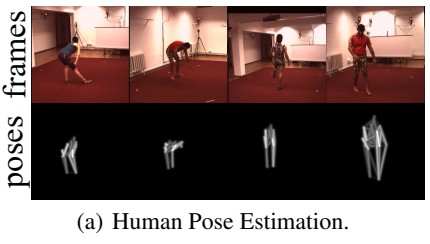

(a) Human Pose Estimation.

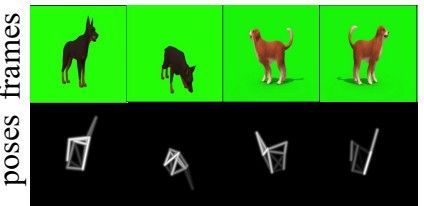

(b) Dog Pose Estimation.

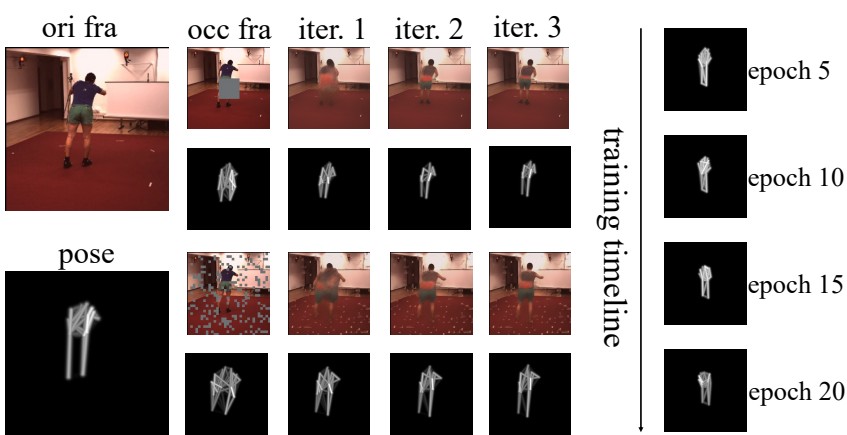

(c) Occluded Human Pose Estimation.

(d) Progressive Pose Priors.

Figure 5: **Visualization Results of estimated poses by PPL on Human3.6m and YouTube Dog Videos.** (a) Pose estimation results on Human3.6m. Row 1 and 2 display eight video frames, while their corresponding estimated poses are shown in Row 3 and 4. The intensity of the lines connecting between any two keypoints indicates their connectivity strengths. (b) Pose estimation results on YouTube Dog videos. Figure conventions follow **(a)**. (c) Pose estimation on occluded frames in Human3.6m. The first column shows the original frame and its estimated pose by PPL. Columns 2-5 show the iterative inference process where the reconstructed frames by PPL (Row 1 and 3) are fed back to itself for estimating poses (Rows 2 and 4) on occluded video frames either using CenterMasking (Row 1 and 2) or RandomMasking (Row 3 and 4). (d) The pose prior evolves as a function of training epochs (from top to bottom).

(Schmidtke et al., 2021), a baseline with human-defined priors, still underperforms compared to our PPL. Consistent with HPE (Yoo & Russakovsky, 2023), this suggests that pre-defined priors are not always optimal, and our PPL is able to learn more representative priors that outperform those manually defined.

**Visualization of Pose Estimation.** We provide the visualization results of pose estimation on Human3.6m (**Figure 5(a)**) and YouTube dog videos (**Figure 5(b)**). The visualization results demonstrate that PPL can learn priors and estimate poses of various object categories, without any

Table 3: **Keypoint Detection Results of our PPL Variants on the Human3.6m dataset.** All results in mean $L2$ errors are normalized by the image resolution of $256 \times 256$. Both keypoint prior (Row 1-2) and Connectivity prior (Row 3-4) can be either pre-defined (Pre.) or randomly initialized (Rand.). During training, the parameters in both the priors can be either frozen (✗) or learnable (✓). The last column (From Mem) shows the result of our default PPL method. Its keypoint prior is initialized from memory (From Mem). Its connectivity prior is randomly initialized (Rand.) and learnable (✓) during training. Best is in bold.

| | | 1 | 2 | 3 | 4 | 5 | 6 | 7 | 8 | 9 | 10 | 11 |
|---|---|---|---|---|---|---|---|---|---|---|---|---|
| Keypoint prior | Initialization | Pre | Pre | Pre | Pre | Pre | Pre | Rand | Rand | Rand | Rand | From mem |
| | Trainable | ✓ | ✓ | ✗ | ✗ | ✓ | ✗ | ✓ | ✗ | ✓ | ✗ | ✗ |
| Connectivity prior | Initialization | Pre | Pre | Pre | Pre | Rand | Rand | Pre | Pre | Rand | Rand | Rand |
| | Trainable | ✓ | ✗ | ✓ | ✗ | ✓ | ✓ | ✓ | ✓ | ✓ | ✓ | ✓ |
| Normalized $L2$ Error | | **2.51** | 2.66 | 2.58 | 2.70 | 2.54 | 2.61 | 2.68 | 2.72 | 2.75 | 2.83 | 2.56 |

external knowledge. For example, in Row 3, Column 2 of **Figure 5(a)**, PPL correctly estimates the bowing pose of a person. In Row 3, Column 5 of **Figure 5(b)**, PPL correctly estimates the pose of a dog lowering down its head. Moreover, we found the quality of estimated poses of dogs is inferior to that of humans. This is primarily due to the significant morphological differences among various dog breeds. Additionally, dogs often perform actions such as turning around, which can lead to significant changes in pose that are difficult to accurately capture by our priors in 2D space.

**Visualization of the Pose Prior Changing with the Training Epochs.** We visualize the progressively learnt pose priors by our PPL as a function of training epochs. **Figure 5(d)** illustrates that the keypoint prior converges to a human shape by the early stage of training (epoch 5). Notably, the learnable keypoints align with the human joints defined in the literature, and the connectivity among keypoints corresponds to the physical connections between body parts. As training continues, the connectivity prior gradually learns the skeletal structure of the human body, with irrelevant links between keypoints diminishing over time (as seen when comparing epochs 15 and 20).

## 4.2 ABLATION STUDIES

**Ablation on Prior Variants.** Here, we investigate how different initializations of connectivity and keypoint priors affect pose estimation and assess whether further refining these priors enhances performance. From **Table 3**, we obtain several key insights: (1) Models with frozen, human-defined priors (Column 4) perform worse than our PPL, indicating that PPL learns more representative priors than those predefined by humans. (2) Refining pre-defined keypoint and connectivity priors (Column 1) outperforms our default PPL, suggesting that PPL can enhance models with human-defined priors through refinement. (3) Interestingly, randomly initializing either keypoint or connectivity priors, followed by refinement during training (Columns 5-9), yields comparable performance to models with human-defined priors. This suggests that human-defined priors may not be necessary for effective pose estimation. (4) Surprisingly, freezing randomly initialized keypoint priors also results in reasonable pose estimation accuracy, though it is still lower than PPL's default performance (Columns 7 and 9). (5) In contrast to (4), freezing random connectivity priors prevents the model from converging, implying that connectivity priors play a more critical role in guiding pose estimations than keypoint priors.

**Ablation on Number & Dimension of Tokens in Each Memory Bank.** In our hierarchical memory, we used 34 memory banks. Here, we analyze the impact of the number of tokens per memory bank and the dimension of each token on PPL's pose estimation performance. From **Figure A1** in **Appendix A3**, we observed that PPL remains robust across different token counts and dimensions, although performance slightly improves with more tokens of higher dimensions. As a result, we fixed 16 tokens per memory bank, each with a dimension of 512, for all experiments.

**Ablation on Number of Keypoints.** We varied the number of keypoints in the pose priors from 4 to 32. The results in **Figure A1** in **Appendix A3** show that pose estimation accuracy improves as the number of keypoints in the prior increases. However, using 32 keypoints offers limited improvement compared to PPL with 16 keypoints for human pose estimation.

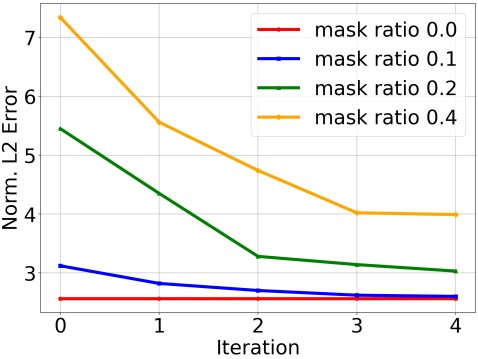 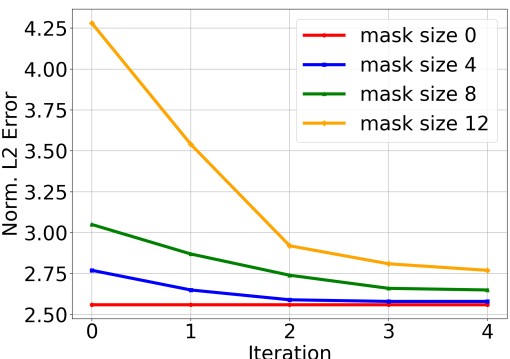

Figure 6: **PPL Results of Keypoint Detection as a Function of Number of Inference Iterations on Images with RandomMasking from Human3.6m.** The "mask ratio" in the legend specifies the masked proportion on the $32 \times 32 = 1024$ image patches.

Figure 7: **PPL Results of Keypoint Detection as a Function of Number of Inference Iterations on Images with CenterMasking from Human3.6m.** The "mask size" in the legend refers to the width and height of the masked region, on 1024 image patches.

### 4.3 POSE ESTIMATION IN OCCLUSION SCENE

To verify the robustness of PPL in occluded scenes, we divide the image into $32 \times 32$ patches and apply two masking techniques: RandomMasking and CenterMasking. In RandomMasking, we randomly mask a certain proportion of image patches, with the proportion ranging from 0.1 to 0.4. In CenterMasking, we mask only the center region of the image, gradually increasing the masking size from $4 \times 4$ to $12 \times 12$ patches.

We explore the effect of occluded areas on PPL. From **Figure 6** and **Figure 7**, we observe that at iteration 0, as the occluded areas increase, overall performance declines with larger occlusions. However, with our iterative inference strategy, PPL effectively infers the missing parts of the poses by utilizing prototypical poses stored in hierarchical memory and the learned priors. Notably, it restores partially occluded poses to reasonably complete full-body poses, leading to a lower $L2$ error, comparable to those without occlusion. This effect is more pronounced when the occlusion area is small or medium.

**Visualization of Pose Estimation with Occlusion.** We present the estimated poses by our PPL for occluded images as a function of the number of inference iterations in **Figure 5(c)**. Across both RandomMasking (top 2 rows) and CenterMasking (bottom 2 rows), with our iterative inference strategy, PPL successfully reconstructs the occluded part of the image after three iterations and meanwhile, predicts reasonable complete full-body poses.

## 5 DISCUSSION

We introduce the challenge of unsupervised prior learning and highlight its significance in pose estimation. To address this, we propose a novel method called Pose Prior Learner (PPL). PPL utilizes a hierarchical memory to store compositional parts of learnable prototypical poses, which are distilled into a general pose prior for any object category. Our experimental results show that PPL requires no additional knowledge and outperforms recent competitive baselines in video pose estimation. Notably, the learned prior proves to be even more effective in pose estimation than methods that rely on human-defined priors. With hierarchical memory and learned priors, PPL can perform iterative inferences and robustly estimate poses in occluded scenes.

Despite outstanding performance in unsupervised pose estimation, PPL has several limitations. For instance, it requires video data captured in static backgrounds. Additionally, PPL learns 2D priors, which makes it difficult to capture real-world 3D postures. Thus, PPL struggles in scenarios where foreground objects involve rotations or significant shape changes. Extending PPL to incorporate 3D priors will be a key focus of our future research.

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

## A1 IMPLEMENTATION DETAILS OF OUR POSE PRIOR LEARNER

We use the Adam optimizer with a learning rate of $10^{-3}$ and a batch size of 64, training for 50 epochs. Unless specified, all images are resized to $256 \times 256$. The learning rate for link weights is scaled by 512 to address the small gradients of SoftPlus near zero. We conduct experiments using the link thickness $\sigma^2 = 5 \times 10^{-4}$ across all benchmark datasets, where we adopt the same definition of $\sigma$ used in (He et al., 2022a). For the hierarchical memory, we use 34 memory banks, each of which contains 16 tokens of dimension 512, for all experiments.

## A2 DATASETS

**Human3.6m** (Ionescu et al., 2013) is a standard benchmark dataset for human pose estimation, consisting of 3.6 million video frames. These frames include both 3D and 2D keypoints and were captured in a controlled studio environment with a static background, featuring various actors. We adhere to the approach outlined in (Zhang et al., 2018; He et al., 2022a), focusing on six activities: direction, discussion, posing, waiting, greeting, and walking. For training, we use subjects 1, 5, 6, 7, 8, and 9, and for testing, we use subject 11.

**Taichi** (Siarohin et al., 2019) consists of 3,049 training videos and 285 test videos featuring individuals performing Tai-Chi, with diverse foreground and background appearances. Following the approach in (Siarohin et al., 2021), we use 5,000 frames for training and 300 frames for testing.

**YouTube dog videos** are videos with green backgrounds collected from YouTube to further qualitatively demonstrate the performance of PPL in non-human species. We used only 20 videos from YouTube and decomposed them into a total of 2000 video frames for training. We use this customized dataset to demonstrate the applicability of PPL in learning pose priors for non-human categories. All images are trained and tested at a resolution of $256 \times 256$. We use 10 keypoints for this category and provide the visualization of the estimated poses on the test videos.

## A3 ABLATION ON MEMORY BANK TOKENS AND KEYPOINT NUMBERS

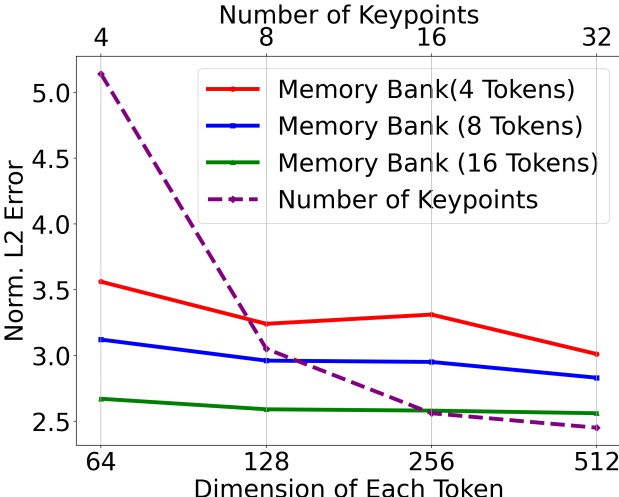

Figure A1: **Ablation of our PPL method on Memory Bank Tokens and Number of Keypoints in Human3.6m.** The upper horizontal axis is the number of keypoints (ranging from 4 to 32) and the lower horizontal axis is the dimension of memory bank tokens (ranging from 64 to 512). The dashed purple line is for ablations on number of keypoints and the solid lines are for ablations on memory bank tokens.

