# OpenReview forum: "Unsupervised Prior Learning: Discovering Categorical Pose Priors from Videos"
_ICLR.cc/2025/Conference — ICLR 2025 Conference Withdrawn Submission_

### Official Review · Reviewer_kNQf · 2024-10-30

**Soundness:** 2
**Presentation:** 2
**Contribution:** 2
**Rating:** 5
**Confidence:** 5

**Summary:**

This paper proposes a framework to learn categorical pose priors from videos, namely PPL. PPL comprises of a cross-frame reconstruction training paradigm conditioned on an reference image for initial keypoints discovery, a connectivity prior similar to AutoLink, and a memory bank trained via vector quantization to store priors. PPL also designs a stage-wise training strategy and several regularizers as losses to stabilize training. PPL iteratively use the learned priors to estimate poses in inference. PPL is an unsupervised learning method and does not require man-made priors. Experimental results show that PPL outperforms previous unsupervised keypoint estimation methods.

**Strengths:**

1. This paper is generally highly completed with detailed explanation of the method, experiments and analysis.

2. The literature review is comprehensive, except for missing some classic work like Monkey-Net [1] and First-Order Models [2]. They are seminal work, should also be included and discussed in the related work rather than just mentioned for its dataset.


[1] Animating Arbitrary Objects via Deep Motion Transfer. CVPR 2019.

[2] First Order Motion Model for Image Animation. NeurIPS 2019.

**Weaknesses:**

1. Lack a clear definition of the problem and motivation.

1.1 Universality? This paper claims that it discovers "a" meaningful pose prior for any object category (L91), and universal prior representations (L93). It conveys the information that the learned prior is universal for any category. However, the designed framework does not have the ability to address different categories, e.g., what if the number of keypoints is not unified across different categories? how to deal with the different connectivity priors for different categories? why the keypoint prior T is a fixed one if it supports different categories? The experiments were also conducted on category-specific datasets like human, dogs. So the paper either overclaims the universality of PPL or makes the problem definition confusing.

1.2 Constraints? Does the motivation of this paper include constraints on keypoint localization and connectivity as stated in L78-79? But
there are existing works for these like AutoLink.

1.3 Mitigating background noise as stated in L109? But it is the contribution from cross-frame reconstuction.

1.4 Comprehensive pose prior without human annotations and domain knowledge as stated in L86-87 and L109? But this paper is still category-specific. It seems not hard to design a category-specific prior, not to mention there is a Prior Tuning mechanism [3].

1.5 The definition of "prior". According to wikipedia, prior knowledge refers to all information about the problem available in addition to the training data [4]. It can be a pretrained model trained from external data, some man-made rules, but cannot be something trained from the data itself. The "prior" proposed in this paper is more like "posterior knowledge". Please scrupulously justify the usage of this term.

2. The contribution is somewhat poor. This paper comprises of cross-frame reconstruction, connectivity prior, and the vector quantization based memory bank.

2.1 The cross-frame reconstruction paradigm for unsupervised pose discovery has been studied over years, originating from Monkey-Net [1] and FOMM [2] in 2019. Tones of recent work also adopts similar framework.

2.2 The connectivity prior in section 3.2 is almost identical to AutoLink. It is hard to say it's the contribution of this paper.

2.3 Use vectory quantization to compress the data to a memory bank is interesting, but similar to weakness 3.2 and 3.3, is it actually useful?

3. Technical concerns.

3.1 Hierarchical memory. This paper mentions several times the memory is "hierarchical", but I cannot find the "hierarchical" characteristic of the memory module. The memory is more like a codebook with quantized vectors, without any "hierarchical" nature.

3.2 It looks like the memory distillation computes the mean pose of the codebook. If so, why not just maintain a running mean of the predicted pose T' as the prior? The pose space is Euclidean and the poses can be averaged. Why take a lot of effort to accomodate a codebook in the framework as well as stagewisely train it while just use the mean of it? The codebook itself might memorize some pose priors but the mean-pooling process makes the keypoint prior T not so informatic.

3.3 Is the memory bank necessary? Why cannot the "prior" knowledge have already been encoded in the $\Phi_{enc}$ via the reconstruction task?

4. Experiments.

4.1 Lack of visualization of results compared with previous work, especially AutoLink.

4.2 The datasets used (Human3.6M, YouTube dogs) have clean or uniform background, which cannot verify the statement in L109 that PPL effectively mitigates background noise.

4.3 Inadequate comparison. Previous work AutoLink was tested on more diverse scenarios including face, fasion, birds, flowers, hands, horses, zebras, etc. What is the performance of this paper against AutoLink under these circumstances?

[1] Animating Arbitrary Objects via Deep Motion Transfer. CVPR 2019.

[2] First Order Motion Model for Image Animation. NeurIPS 2019.

[3] Efficient, Self-Supervised Human Pose Estimation with Inductive Prior Tuning. ICCV 2023.

[4] Wikipedia: Prior knowledge for pattern recognition. https://en.wikipedia.org/wiki/Prior_knowledge_for_pattern_recognition

**Questions:**

1. Evaluation protocol. L369, why not use previous evaluation protocol? Why employing a minimum flow algorithm is a more accurate evaluation protocol than the one used in previous work? What are the comparison results when using previous protocol?

2. Some presentation issues.

2.1 The sentence in L130-131 is contradictory with the fact of cross-frame reconstruction methods like BKind which learn from video frames. Different frames naturally eliminate the response from background which has been verified by plenty of work.

2.2 Figure 2. The thumbnail of $I_{recon}$ should use $I$ or a reconstructed $I$, but cannot be $I_{ref}$.

---

### Official Review · Reviewer_T524 · 2024-10-31

**Soundness:** 2
**Presentation:** 3
**Contribution:** 2
**Rating:** 5
**Confidence:** 4

**Summary:**

The authors propose a method for learning pose priors, specifically, keypoints and their connectivity, from videos. Towards this, they propose the Pose Prior Learning framework which does not utilize any human intervention or annotations (unsupervised). Furthermore, PPL employs a hierarchical memory to store and distill a general pose prior from these video frames and this memory-based supposedly allows the model to refine its pose estimations iteratively. The results are compared with baseline models on human datasets (H3.6 and Taichi) and visualized for animal (YouTube dog videos) datasets.

**Strengths:**

1. An unsupervised approach does not require annotations and is thus useful for learning priors, in this case, pose priors
2. The proposed pipeline should prevent the background information from flowing through the image encoder and so it acts as a bottleneck which can be effective in disentangling pose from the image
3. The hierarchical memory system aggregates prototypical poses which allows PPL to distill robust pose priors and improve pose estimation in challenging scenarios like occlusion
4. The authors show that PPL outperforms other methods on human pose estimation benchmarks - H3.6 and Taichi which establish the effectiveness of the proposed approach on human datasets

**Weaknesses:**

1. The authors note that the proposed method works on any object category. However, the quantitative results are only shown on human datasets. There are plenty of animal datasets [1, 2] already publicly available that the authors could evaluate PPL on. Furthermore, there are available methods for animal pose estimation/key points detection (supervised and unsupervised) and the authors do not provide comparisons on them [3, 4, 5]. There is no justification provided for the absence of quantitative evaluations. This raises concerns regarding the generalizability of the method on different object categories other than humans.
2. The assumption of a static background with changing poses is not always feasible and thus the method has limited practical applications. One workaround could be segmenting out the background and just using the foreground.
3. In spite of using video frames, the model does not make use of temporal information and thus results on video data would have limited performance. Though the authors show results on video datasets, there is no evaluation of the model’s performance for consistent keypoint prediction across frames.

[1] APT-36K: A Large-scale Benchmark for Animal Pose Estimation and Tracking
[2] Cross-Domain Adaptation for Animal Pose Estimation
[3] From Synthetic to Real: Unsupervised Domain Adaptation for Animal Pose Estimation
[4] A Horse with no Labels: Self-Supervised Horse Pose Estimation from Unlabelled Images and Synthetic Prior
[5] Unsupervised Learning of Object Landmarks through Conditional Image Generation

**Questions:**

1. Are the key points in T known beforehand? For example, do the authors assume that keypoint P1 belongs to the head, and so on? If not, why can the authors not assume this information and learn a prior on this which should be easier? Can the authors clarify their keypoint initialization process and discuss the trade-offs between using pre-defined keypoint semantics versus learning them from scratch?
2. Why is there a need to learn connectivity information? The connectivity is fixed for humans and animals. For example, in L189-191 the probability of a hand connecting a foot should be 0. The authors do show an ablation in Table 3 regarding this but there is no explanation as to why the authors did not choose to start from a pre-defined connectivity at first.
3. What is the need for collecting a dog dataset when there are available animal datasets (See weaknesses)?
4. Using the same variable ‘m’ to represent memory banks and tokens is ambiguous. Please use a different variable
5. Will the YouTube dog videos dataset be released? Does the dataset come with annotations? If yes, do the authors manually annotate the dataset or use some pre-trained network to get the annotations?
6. Can the proposed approach be extended to rigid objects such as a ball or a vehicle?

---

### Official Review · Reviewer_UPa3 · 2024-11-03

**Soundness:** 2
**Presentation:** 2
**Contribution:** 2
**Rating:** 3
**Confidence:** 4

**Summary:**

The paper presents Pose Prior Learner (PPL) for learning pose priors from video data to improve pose estimation without human annotations. By employing a hierarchical memory that aggregates compositional parts of prototypical poses, PPL derives a general pose prior that enhances pose accuracy across different object categories, including humans and animals. This paper demonstrates superior performance over traditional methods, as it mitigates background noise, requires no human intervention, and even surpasses models that rely on predefined priors, making it a promising tool for robust, explainable pose estimation in diverse visual contexts.

**Strengths:**

The authors propose a unsupervised approach to prior learning in pose estimation. The proposed method Pose Prior Learner (PPL) combines self-supervised learning and compositional memory. Experiments on multiple datasets have demonstrated that this method can alleviate the problems caused by background noise and occlusion. The paper is well-structured, with clear illustrations that elucidate the architecture and iterative inference process.

**Weaknesses:**

The paper presents a compelling approach, but there are several areas where it could be strengthened.
1. The definition of each variable should be more clear. Eg., the definition of m in eq.4.
2. The author has many overly long sentences and defines many variables throughout the article, which makes the whole article difficult to read. Eg. $T$, $T'$ and $T'_{recon}$.
3. In L187, why are pixel coordinates defined as $P_i$ $\in$ [-1,1] $\times$ [-1,1] instead of two-dimensional coordinates?.
4.Why to reconstruction the frame? Even if the background is consist, how to keep the same texture and inpainting the unknow area? Besides, the reconstruction part is not very clear. Without the details of methods, loss functions and the optimization goals.
5. The author claims in the main contribution that this paper establishes evaluation metrics and benchmarks, but I did not see the metrics definition or justification of evaluation metrics. In my opinion, a new evaluation metrics and benchmarks should be proved by extensive and rigorous experimental evidence. Explain why former metrics and benchmarks are useless are also important.

**Questions:**

I have listed the questions in weakness part and hope the response of authors.

---

### Official Review · Reviewer_3akK · 2024-11-03

**Soundness:** 1
**Presentation:** 1
**Contribution:** 1
**Rating:** 5
**Confidence:** 3

**Summary:**

The paper introduces Pose Prior Learner (PPL), a method for unsupervised pose estimation that learns pose priors from unannotated videos. PPL’s hierarchical memory architecture captures prototypical poses, which are distilled into a general pose prior to improve pose estimation, particularly for occluded images.

**Strengths:**

PPL effectively applies hierarchical memory to unsupervised pose estimation, storing and refining prototypical poses from video data, which enhances pose estimation accuracy without human annotations.

The iterative inference process enables the model to perform robustly even in occluded settings, as demonstrated across multiple datasets.

Visualization and ablation studies provide clarity on PPL’s memory structure and the effects of learned priors, supporting the approach’s design choices.

**Weaknesses:**

The paper overclaims the challenge of prior learning in Contribution 1 in Lines 105-106. First, as stated in Lines 68-70, previous works also learn the pose priors and the authors agreed on that part, so it is not this paper that introduces the challenge of unsupervised prior learning on pose estimation tasks. Second, compared to previous works,  it is unclear about the contribution this paper has made based on previous pose prior learning methods.

The paper overclaims that the proposed PPL method does not require any domain knowledge from humans in Contribution 2. The proposed method estimates pose from specific categories, which need human knowledge to classify objects into different categories.

There are many unclear expressions in the paper:

what are the red lines and black lines in Figure 2? Why the red line can be pointed to the frame $I$?

In Line 186, how is the $(T, W)$ determined by different categories?

Line 114, We -> we

Line 218, incorrect ";"

Line 253, it -> them

**Questions:**

See weakness.

---

### Note · Authors · 2024-11-30

I have read and agree with the venue's withdrawal policy on behalf of myself and my co-authors.